# Is Hematopoietic Stem Cell Transplantation Required to Unleash the Full Potential of Immunotherapy in Acute Myeloid Leukemia?

**DOI:** 10.3390/jcm9020554

**Published:** 2020-02-18

**Authors:** Edward Abadir, Robin E. Gasiorowski, Pablo A. Silveira, Stephen Larsen, Georgina J. Clark

**Affiliations:** 1Dendritic Cell Research, ANZAC Research Institute, Concord 2139, NSW, Australia; pablo.silveira@sydney.edu.au; 2Institute of Haematology, Royal Prince Alfred Hospital, Camperdown 2050, NSW, Australia; stephen.larsen@health.nsw.gov.au; 3The University of Sydney, Camperdown 2039, NSW, Australia; robin.gasiorowski@health.nsw.gov.au; 4Department of Haematology, Concord Repatriation and General Hospital, Concord 2039, NSW, Australia

**Keywords:** AML immunotherapy, hematopoietic stem cell toxicity, Chimeric Antigen Receptor (CAR) T cells, antibody drug conjugates

## Abstract

From monoclonal antibodies (mAbs) to Chimeric Antigen Receptor (CAR) T cells, immunotherapies have enhanced the efficacy of treatments against B cell malignancies. The same has not been true for Acute Myeloid Leukemia (AML). Hematologic toxicity has limited the potential of modern immunotherapies for AML at preclinical and clinical levels. Gemtuzumab Ozogamicin has demonstrated hematologic toxicity, but the challenge of preserving normal hematopoiesis has become more apparent with the development of increasingly potent immunotherapies. To date, no single surface molecule has been identified that is able to differentiate AML from Hematopoietic Stem and Progenitor Cells (HSPC). Attempts have been made to spare hematopoiesis by targeting molecules expressed only on later myeloid progenitors as well as AML or using toxins that selectively kill AML over HSPC. Other strategies include targeting aberrantly expressed lymphoid molecules or only targeting monocyte-associated proteins in AML with monocytic differentiation. Recently, some groups have accepted that stem cell transplantation is required to access potent AML immunotherapy and envision it as a rescue to avoid severe hematologic toxicity. Whether it will ever be possible to differentiate AML from HSPC using surface molecules is unclear. Unless true specific AML surface targets are discovered, stem cell transplantation could be required to harness the true potential of immunotherapy in AML.

## 1. Introduction

Until recently, therapeutic options for Acute Myeloid Leukemia (AML) had changed very little. In the last decade, there has been a substantial increase in knowledge of the molecular landscape of AML, which has led to several new targeted therapies [1]. However, many of these molecular therapies have limited duration of action when not combined with conventional chemotherapy [2]. Since the FDA approval of Rituximab in 1997, immunotherapy has become an increasingly important part of the management of hematologic malignancies [3]. Unconjugated antibodies, Antibody Drug Conjugates (ADC), Bispecific T Cell Engagers (BiTEs), and Chimeric Antigen Receptor (CAR) T cells are all now part of accepted therapy.

More potent forms of immunotherapy, such as Chimeric Antigen Receptor (CAR) T cells, lead to target cell aplasia, which is tolerable in B cell malignancies [4,5,6]. Indeed, nearly all the benefits of targeted immunotherapy are in the setting of B cell malignancies. Aside from Gemtuzumab Ozogamicin (GO), no immunotherapies have been accepted for hematologic malignancies of myeloid origin [7]. The limited efficacy in AML to date highlights the challenges presented.

Most surface molecules expressed on AML are also expressed on Hematopoietic Stem and Progenitor Cells (HSPC), and potent immunotherapies against traditional AML molecules (CD33 and CD123) have led to hematologic toxicity, both in preclinical models and in clinical trials [8,9,10]. An additional issue is that the complex subclonal architecture and plasticity of surface molecules makes tumor escape a significant concern [11,12,13,14]. Despite these challenges, new immunotherapies against AML continue to be tested. Allogeneic Hematopoietic Stem Cell Transplantation (allo-HSCT) is an established cellular therapy in AML, but the toxicity and limited efficacy, especially in high risk patients, allows for potential improvement. In this procedure, patients undergo conditioning chemotherapy with or without radiotherapy followed by transfusion of donor hematopoietic cells. Immunosuppression is required after the transplant to reduce the chance of Graft Versus Host Disease (GVHD). Allo-HSCT has been shown to reduce the relapse rate of AML and is the only potentially curative therapy in those with refractory disease [15,16]. The decision on whether to offer an allo-HSCT is complex and considers patient fitness, risk of AML relapse, and availability of donors. Generally, if a suitable well-matched donor is available allo-HSCT is recommended for those with intermediate or adverse risk disease [15]. The major limitation is that most patients are not fit for the procedure due to their age at diagnosis and other comorbidities. Autologous HSCT is an alternative to allo-HSCT that is associated with reduced toxicity, but is only effective in patients without high risk disease and is not widely utilized in all jurisdictions [17].

While the majority of the emerging anti-AML immunotherapies seek to control disease without allo-HSCT by sparing HSPC (Figure 1), others look to targeted immunotherapy to act as a component of allo-HSCT and accept that depleting HSPC is required to control AML (Figure 2).

Graft Versus Leukemia (GVL) in allo-HSCT provides evidence that immune mechanisms can reduce relapse risk in AML. The evidence of GVL was seen in preclinical models and confirmed clinically with the observations of reduced relapse rates in patients who develop GVHD, and an increased relapse rate in patients who receive T cell depleted grafts [18,19]. The immune mechanism of GVL is complex; incompletely understood; and involves T, NK, and antigen-presenting cells with likely multiple leukemia antigens [20]. This contrasts to current immunotherapies, which primarily target a single molecule. Immunotherapy can often play a complementary role with allo-HSCT, acting as bridge to transplant. This is seen in B-cell Acute Lymphoblastic Leukemia (B-ALL), where the ADC, Inotuzumab Ozogamicin, and Bispecific T Cell Engagers (BiTE), Blinatumomab, have been shown to eliminate leukemic cells capable of achieving Minimal Residual Disease (MRD) negativity prior to allo-HSCT [21,22]. The emergence of immunotherapies altering the treatment landscape in regard to allo-HSCT can be seen in the field of B-ALL. CD19 CAR T cells are capable of inducing sustained Complete Remission (CR), thus potentially minimizing the role of allo-HSCT in children and young adults with relapsed ALL [23]. Despite the emergence of CAR T cells in B-ALL, it is unclear if the need for allo-HSCT will be reduced over time or if these therapies will play complementary roles. Given that relapse occurs post CD19 CAR T cell therapy, the role of consolidation with an allo-HSCT is not currently established [24,25].

Optimal disease control prior to allo-HSCT in AML is an important part of reducing relapse risk. Patients with refractory disease going into to transplant have a much higher risk of relapse compared to those in CR [26,27]. Those who are MRD-positive at the time of transplant have worse outcomes [28,29,30]. Even though MRD positivity is associated with a high relapse rate post allo-HSCT, these patients may still benefit from transplant [31]. Immunotherapy in AML may contribute as a bridge to transplant, as has been established in B-ALL [21,22]. There are already emerging strategies reported of adding targeted AML agents to conventional conditioning regimens, which can reduce the relapse risk post-transplant in high-risk patients with AML [32]. As AML and HSPC share antigens, it is possible that the step of disease depletion with immunotherapy prior to allo-HSCT can also act as a form of conditioning. This application would have the advantage of depleting residual disease and reducing nonhematologic toxicity by potentially replacing nonspecific conditioning agents that eliminate host HSPC. The disadvantage of this strategy is the requirement of an allo-HSCT.

Allo-HSCT carries significant limitations. The median age of diagnosis of AML is 72, which prevents a majority of patients from accessing allo-HSCT due to treatment related toxicity [33]. Allo-HSCT requires chemotherapeutic agents as conditioning to suppress the host’s immune system while removing residual AML and HSPC. These factors are critical to allow for the safe engraftment of donor cells. The development of nonmyeloablative and Reduced Intensity Conditioning (RIC) regimens has allowed more patients to benefit from allo-HSCT by reducing the treatment-related mortality [34]. These developments have made transplantation in selected patients over the age of 65 and increasingly over the age of 70 possible, but not without significant morbidity and mortality [35,36]. In attempts to broaden the benefits of allo-HSCT, there is an emerging effort to reduce treatment related mortality from allo-HSCT by replacing traditional conditioning immunotherapy [37]. Clinical trials (NCT02963064) have been established in children with severe combined immunodeficiency receiving chemotherapy and radiation free conditioning with a CD117 antibody, with early reports demonstrating partial engraftment [38]. An antibody-based conditioning strategy against CD117 has been shown to be effective in eliminating myelodysplastic syndrome HSPC and allowing for engraftment of healthy HSPC preclinically [39]. This emerging field will likely converge with AML immunotherapy due to shared targets.

### Is Disease Control Linked to Hematologic Toxicity? Implications from Targeting CD33

CD33/Siglec-3 is a protein from the immunoglobulin superfamily, with expression that ranges from myeloid progenitors to mature myeloid cells [40]. CD33 is expressed in 90–95% of AML [41,42]. Positivity is often defined by a specific median fluorescent intensity ratio by flow cytometry, using either an external control or an internal negative control such as lymphocytes [42,43,44]. Despite the vast majority of cases meeting positive median fluorescent intensity ratio criteria, a substantial proportion of cases have less than 70% AML cells expressing CD33 when examined on an individual cell level and this impacts the efficacy of immunotherapy [44]. It is accepted that CD33 is expressed on myeloid progenitors but there have been conflicting reports on the expression on Hematopoietic Stem Cells (HSC) [45,46]. Functional studies demonstrating CD33+ cells are capable of serial engraftment in immunodeficient mice suggests its expression on HSC [45].

Gemtuzumab Ozogamicin (GO) is an ADC using a calicheamicin warhead and the only currently listed form of immunotherapy for AML. Its history with regulatory authorities is long and complex. GO was given accelerated approval in 2000 by the FDA only for it to be withdrawn in 2010 [47,48]. The results of the ALFA-0701 study and a meta-analysis demonstrating improved survival in low to intermediate risk patients led to a subsequent GO approval by the FDA in 2017 for the treatment of AML in the upfront setting with a fractionated dosing schedule [48,49,50,51]. While the increased rates of sinusoidal obstruction syndrome (SOS), especially with higher doses, are the best known adverse effect of GO, hematological toxicity has been a feature of GO across several studies [48]. The results from the ALFA-0701 study using the recommended fractionated dosing schedule during induction showed prolonged neutropenia after consolidation phases as well as prolonged and persistent thrombocytopenia after induction [50]. When used outside of intensive chemotherapy, GO was associated with universal pancytopenia but did increase survival compared to best supportive care [52]. While GO has tolerable hematologic toxicity and is capable of benefiting some patients with AML, there is still a very large proportion of patients who do not benefit from it. More potent forms of immunotherapy have been targeted against CD33, and while these may have theoretical advantages in efficacy, they come at the price of increased hematologic toxicity.

Vadastuximab talirine (VT), SGN33a, is an anti-CD33 ADC utilizing a PyrroloBenzodiazepine Dimer (PBD) toxin. VT has a number of theoretical advantages over GO, including uniform drug loading, effectiveness in multidrug resistance positive cell lines, and activity against AML with adverse cytogenetics [53]. VT has displayed impressive CR rates, including MRD negativity when used in the upfront setting as well as a higher rate of response when combined with hypomethylating agents compared to historical controls [54,55]. The potential for significant myelosuppression by targeting CD33 was supported by a dose limiting toxicity (60 ug/kg) of hypocellular marrow in a phase I trial for VT [10]. The outcomes of older patients with newly diagnosed AML treated with VT monotherapy revealed that prolonged neutropenia (median 6.1 weeks) and thrombocytopenia (median 5.1 weeks) were common in those who achieved CR or CR with incomplete recovery [56]. A phase 3 trial of hypomethylating agents +/- VT was suspended after excess deaths due to infection were observed in the VT arm [57]. With increasingly potent therapeutics available, avoiding hematologic toxicity while remaining effective against AML may require an allo-HSCT as a rescue.

## 2. Strategies Trying to Mitigate Hematologic Toxicity

### 2.1. Alternate Surface Targets

While there remains several ongoing strategies to target CD33 in AML, the challenges in the development of VT highlight some of the difficulties with this target [58]. Alternate targets with reduced expression across HSPC have become appealing given the potential for reduced hematologic toxicity.

C-Type Lectin-Like Molecule 1 (CLL-1, CD371) is a surface molecule found on AML and committed myeloid progenitors [59]. A CLL-1xCD3 BiTE was broadly effective against a range of AML cell lines [60]. It induced neutropenia that recovered by day 22 in cynomolgus monkeys. This finding confirms the potential for neutropenia, but the recovery suggests a potential to spare HSC. An ADC targeting CLL-1 was effective against in AML cell lines in vitro and in vivo [61]. The ADC significantly reduces granulocytic Colony Forming Units (CFU) while sparing erythroid CFUs. This ADC does not reduce the number of human CD45^+^ cells in a humanized mouse model. A different ADC against CLL-1 with a PBD toxin depleted AML cell lines and primary AML in vitro and prolonged survival of mice xenografted with an AML cell line [62]. Limited preclinical hematologic toxicity was demonstrated, as the ADC caused reversible neutropenia in cynomolgus monkeys without a reduction in platelets or lymphocytes. These results suggest that an anti-CLL-1 immunotherapy may lead to transient neutropenia with a predictable recovery. While neutropenia is a significant toxicity, it is common amongst AML therapeutics and the consistent recovery across different therapeutics suggests that HSC and early progenitors are unlikely to be targeted by an anti-CLL-1 therapeutic.

T cell immunoglobulin mucin-3 (TIM-3, CD366) has emerged as a potential selective target against AML while sparing normal hematopoiesis. TIM-3 is expressed on AML blasts and is upregulated on T cells of AML patients [63,64]. TIM-3 appears to play a role in Leukemic Stem Cell (LSC) establishment, as TIM-3 antibodies can block engraftment of AML in immunocompromised mice [65]. Within HSPC, only a subset of Granulocyte/Macrophage Progenitors express TIM-3 when assessed by flow cytometry [65]. A TIM-3 antibody was able to deplete TIM-3^+^ monocytes in a humanized mouse model but did not impair the development of B or myeloid cells [65]. An early report of a TIM-3 antibody in combination with decitabine in hypomethylating agent naive patients has demonstrated limited toxicity but also limited efficacy to date with a CR rate of 14% (2/14) [66]. A CAR T cell has been developed that selectively kills targets expressing CD13 and TIM-3 [67]. This strategy still depleted HSPC but not as much as targeting CAR T cells directed against only CD13. Weather a limited long-term depletion of HSPC has clinically significant impacts is unknown.

Natural Killer Group 2D (NKG2D) is a surface receptor that is expressed normally on NK and CD8^+^ T cells, but is also expressed on stressed and malignant cells, including the majority of AML [68,69]. Its absence on resting healthy tissues makes it an attractive target. A case report from a Phase I/II trial (NCT03018405) with a patient who had relapsed FLT3^+^ AML who underwent NKG2D CAR T cell therapy demonstrates initial promise [70]. The patient achieved a complete morphological response but had evidence of clonal evolution with a new *IDH2* mutation. Despite this finding, he had normal trilineage hematopoiesis, providing evidence of the lack of HSPC expression of NKG2D. He subsequently underwent allo-HSCT, leading to CR with normal molecular studies. It is unclear if AML subclones may have variable NKG2D expression; this case highlights the challenges in targeting a heterogeneous disease like AML with a single surface molecule. More data are required to determine if NKG2D targeted therapy can be a standalone therapy or a bridge to transplant.

CD70 is the ligand of CD27, and this interaction helps to regulate lymphocyte and HSPC activity [71]. CD70 has minimal expression on healthy HSPC and is substantially upregulated in AML as well as other malignancies [72,73]. The blockade of CD70 with a monoclonal antibody inhibits self-renewal of AML and LSC while extending survival in a xenograft model of primary AML [72]. Hypomethylating agents have been shown to induce CD70 expression further on AML, thus suggesting synergistic potential with CD70 antibodies, and a phase II trial using this combination in previously untreated AML is underway (NCT04023526) [74]. What remains unclear is if blocking the CD70/CD27 axis will affect hematopoiesis even if the HSPC themselves do not express CD70.

While many of the targets to date have been expressed on a majority of AML, there have also been efforts to target surface proteins that are aberrantly expressed on AML even if they constitute a minority of cases. CD7 is a cell surface glycoprotein that is normally expressed on T and NK cells as well as their progenitors [75]. CD7 is expressed in 30% of AML cases [76]. The lack of CD7 expression on HSPC reduces the chance for severe hematologic toxicity resulting from targeting CD7. A CAR T cell directed against CD7 depleted AML cell lines in vitro and in vivo but did not reduce CFU formation from normal cord blood [77]. The possibility of CAR T cell fratricide is reduced by editing the CD7 gene on the CAR T cells so they do not display their target [78]. This strategy would only be suitable for a minority of patients with AML, and the impact of immunosuppression from potentially long-term healthy T cell depletion is unclear.

Another method of target selection that may reduce hematologic toxicity is to selectively target markers specific for monocytic differentiation. Leukocyte immunoglobulin-like receptor-B 4 (LILRB4, CD85k, ILT3) is expressed from promonocytes to mature monocytes [79]. LILRB4 is also expressed on AML with monocytic differentiation i.e., M4 and M5 by FAB classification [80]. A CAR T cell directed against LILRB4 demonstrated efficacy against M5 AML cell lines and primary M5 AML in vitro as well as a M5 AML cell line in vivo [81]. It did not reduce CFU numbers and in a humanized mouse model did not deplete CD34^+^, CD33^+^, or CD19^+^ cells. M5 AML was chosen as the target, as M4 AML did not uniformly express LILRB4. The targeting of M5 AML limits the number of potential patients who may benefit as AML with monocytic differentiation accounts for only 5-10% overall, though the proportion is up to 40% in children [82,83].

CD300f is another surface target expressed across AML, mature myeloid cells, and HSPC [84,85]. There are seven isoforms of CD300f described, and it has been shown that exon 4 expression of the protein is selectively upregulated in AML with monocytic differentiation compared to healthy HSPC [85]. In addition, a monoclonal antibody can enact a conformational change of CD300f allowing a second antibody to bind to AML with monocytic differentiation at high affinity, but not to HSPC. CD300f is a promising target in AML with monocytic differentiation, but more work remains to develop a comprehensive strategy to utilize the selective expression and spare HSPC from future therapeutics.

### 2.2. Intracellular Targets

While AML-specific surface targets have been difficult to validate, intracellular targets have been described that are substantially upregulated in AML compared to normal populations. The best known upregulated AML antigens are WT1 and PRAME, which are found in the majority of AML samples [86]. Despite their suitability for targeting, the difficulty accessing intracellular antigens has limited their development in immunotherapy.

Dendritic cell (DC) vaccines have been the most common strategy to target AML intracellular antigens, with a wide range of manufacturing strategies and targets explored [87]. DCs loaded with AML antigens are able to stimulate T and NK cells to induce an antileukemia effect [88]. The main advantage of DC vaccination is the demonstrated favorable safety profile, which is especially important given the advanced average age at presentation [88]. The main barriers to entering routine clinical practice are the variability across different strategies and limited efficacy [87]. The type of DC used may be one of the limiting factors in vaccination efficacy. Most studies have used monocyte-derived DCs, which are less efficient at antigen presentation and the generation of T cell responses [89,90]. Using blood DC has the potential to overcome some of the limitations in DC vaccine efficacy [91].

An alternate way to target intracellular antigens and possibly increase the efficacy of treatment over DC vaccination is to use T-cell receptor (TCR)-engineered T cells. Engineered T cells can be designed to recognize intracellular antigens expressed by MHC on AML [92]. CD8^+^ T cells reactive against WT1 on HLA-A*02:01 are functional in vitro, are home to the bone marrow of patients, and persist after reinfusion [93]. Persistence of engineered T cells against WT1 expressed on HLA-A*24:02 has been demonstrated in an MDS setting [94]. The limited patient numbers published make the assessment of clinical efficacy difficult. A central limitation with this approach is that engineered T cells must be made to recognize a specific HLA allele, thereby restricting utility to only those patients who express the targeted HLA.

### 2.3. Alternate Effector

A separate strategy to reduce hematologic toxicity is to use a toxin as part of an ADC that demonstrates increased efficacy against AML compared to HSPC. It may be possible to find an effective therapeutic window where an ADC can deplete AML while sparing HSPC. IMGN632 is an ADC that targets CD123 and uses an indolinobenzodiazepine pseudodimer alkylator payload that demonstrates activity against AML cell lines and primary AML in vitro as well as AML cell lines in vivo [95]. IMGN632 had an increased therapeutic window when compared to GO with myeloid progenitor and AML CFU assays. The antibody clone of IMGN632, G4723A, was also conjugated to an alternate more potent toxin that had similar IC90 values between AML and myeloid progenitors. The alternate ADC was >40x more toxic to myeloid progenitors than IMGN632, which demonstrates the importance of the toxin over the target in providing an extended therapeutic window.

## 3. Strategies Incorporating Allogeneic Hematopoietic Stem Cell Transplantation

Due to the difficulties in avoiding HSPC depletion with anti-AML immunotherapies, an alternative strategy would be to incorporate these therapies into allo-HSCT, which is an established therapy for AML. Attempts to target CD123 with CAR T cells demonstrate how immunotherapies against AML may be incorporated into a transplant model. CAR T cells have shown preclinical in vitro and in vivo efficacy against AML, but there are conflicting reports on the potential for toxicity against HSPC. Some studies demonstrated no effect on myeloid progenitors, while others displayed complete myeloablation in humanized mouse models [8,96]. Depleting CAR T cells after a fixed period is a potential strategy to circumvent destruction of donor HSPC during allo-HSCT. CAR T cells can be depleted by targeting their natural or engineered surface molecules. CAR T elimination has been demonstrated with alemtuzumab, or rituximab when CD20 is introduced during CAR T cell manufacturing [97]. A functional proof of CAR T potential in allo-HSCT was demonstrated by using CD123 CAR T cells to deplete human cells in a mouse sequentially engrafted with human T cell deplete bone marrow and a human AML cell line [97]. This was followed by rituximab administration to remove the CAR T cells before a subsequent sex mismatched human donor graft [97]. “Bio-degradable” CD123 CAR T cells, which lack the ability for indefinite self-perpetuation, were administered to patients as part of a Phase I trial to exclude toxicity [92]. These CAR T cells were well tolerated but did not display efficacy against AML [92]. The correlation between persistence of CD19 CAR T cells and continued response in B-ALL suggests that a prolonged presence of CAR T cells may be required for efficacy; this may limit the effectiveness of short-term CAR T cells. There are currently clinical trials using CD123 CAR T cells in AML that do (NCT03766126) and do not (NCT04109482) require a nominated transplant donor prior to enrolling, as a precaution in the event of bone marrow aplasia.

An innovative way to allow anti-AML CAR T cells to persist is to remove a common target from HSPC. This strategy has been explored by developing CAR T cells against CD33 and then removing the CD33 from an allo-HSCT donor graft [98]. CD33-depleted myeloid cells did not display any significant developmental or functional differences, and they were able to be manufactured from both human and nonhuman primates HSPC [98]. The CD33 CAR T cells were able to deplete an AML cell line in CD33 KO humanized mice without reducing normal myelopoiesis. While this strategy may circumvent many theoretical problems with immunotherapy in AML, practical issues including cost may become limiting. CAR T persistence would likely increase efficacy but the possibility of antigen escape remains, as seen with CD19^-^ relapses in ALL [99]. To combat this issue, immunotherapies targeting multiple antigens have been developed.

A compound CAR T cell targeting CD33 and CD123 has been developed that demonstrates independent activity against cells expressing either molecule [100]. These CAR T cells were effective in vitro against AML cells lines and were able to prolong survival of xenografted mice with AML cell lines displaying CD33, CD123, or both CD33 and CD123. To enhance safety and reduce the chance of prolonged bone marrow aplasia, the compound CAR T cells were able to be depleted in vivo using alemtuzumab. Results from Phase 1 trial (NCT03795779) of a compound CLL-1 CD33 CAR T cell have been presented. The first patient reported conversion from active disease to MRD negative with pancytopenia; this was followed by a successful matched sibling transplant [101]. A second case presentation with this construct demonstrated ablation of AML and pancytopenia as a result of CAR T cell administration followed by a allo-HSCT [101]. Given the limited efficacy detailed in anti-AML CAR T cell case reports and early phase trial results, compound targeting may represent a significant improvement on sustained efficacy against AML [92]. By incorporating an increasing number of shared targets, this strategy makes it even more likely to have significant hematological toxicity without allo-HSCT rescue.

In addition to CAR T cells, there have been reports of ADCs against CD45 and CD117 to reduce AML and conditions for an allo-HSCT [102]. These antibodies use the ribosomal inhibiting toxin amanitin and are effective at prolonging survival in mice bearing AML xenografts. Both the CD45 and CD117 amanitin ADC depleted human and nonhuman primate HSPC to confirm the potential utility in conditioning [103,104]. These results are still preliminary but offer a possible pathway to AML immunotherapy incorporating allo-HSCT that avoids the complexities of CAR T cell generation.

## 4. Conclusions

Effective immunotherapy for most patients with AML remains elusive. The experience with GO suggests that immunotherapy may be able to play a role in AML treatment, but the clinical trials with VT illustrate the difficulty in managing more potent therapeutics against AML with the associated increase in hematologic toxicity. Unless a true leukemia specific epitope is found that is consistently expressed across at an individual cell level, then targeting single molecules may continue to show limited efficacy. Multiantigen targeting increases the risks of hematologic toxicity but would likely reduce the chance of tumor escape.

Requiring an allo-HSCT is a major disadvantage of any potential future therapy, as most patients with AML are ineligible for allo-HSCT. The concept of bridging to allo-HSCT may change with AML immunotherapy. In other hematologic malignancies, bridging therapy occurs before conditioning; therefore, the treatment is naturally longer. In AML therapy, it may be possible to combine bridging with conditioning, which may reduce the toxicity of the individual elements combined. Another potential benefit of incorporating immunotherapy in allo-HSCT conditioning for AML is that these therapies may replace traditional conditioning agents. The consistent presentation of AML targets on HSPC may allow for an extension immunotherapy to conditioning in other diseases, including myelodysplastic syndromes as well as inherited disorders of hematopoiesis and immunity that respond poorly to traditional conditioning agents.

The design and trials of emerging AML immunotherapy must be carefully considered to have a chance of success, given the high morbidity and mortality associated with the disease itself. Tolerable and effective immunotherapy that can reduce the need for allo-HSCT would constitute a major change in the treatment of AML, but to date this has been elusive. Finally, the alternative of enhancing the efficacy while reducing the toxicity of allo-HSCT with immunotherapy would constitute substantial progress in finding better outcomes for patients with AML.

## Figures and Tables

**Figure 1 jcm-09-00554-f001:**
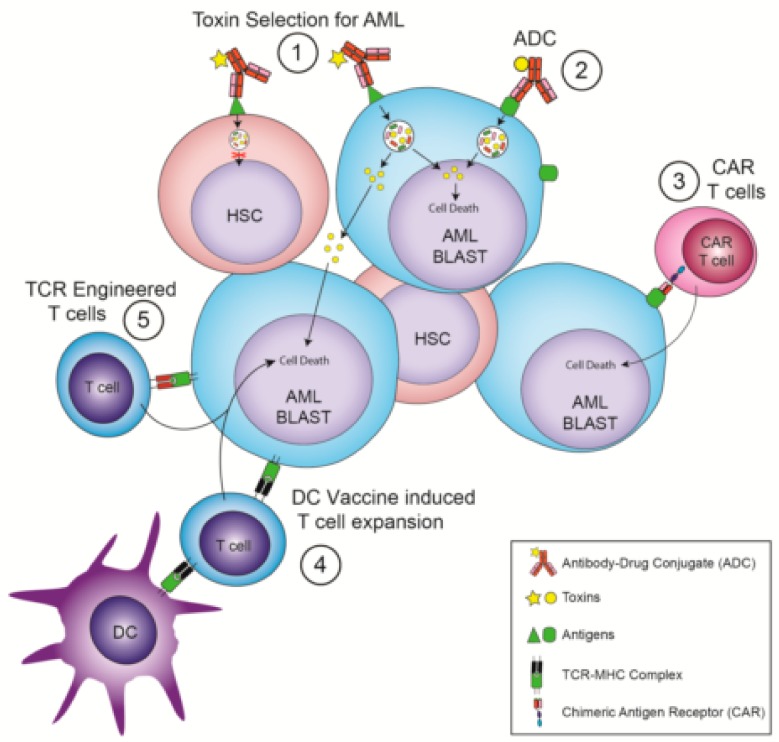
Immunotherapies that avoid the need for hematopoietic stem cell transplantation. (1) Antibody Drug Conjugates (ADC) that recognize Hematopoietic Stem Cells (HSC) and Acute Myeloid Leukemia (AML) but whose toxin selectively kills AML cells. (2) ADC that recognize AML-specific surface targets. (3) Chimeric Antigen Receptor (CAR) T cells that recognize AML-specific surface targets. (4) Dendritic cell (DC) vaccination leading to a T cell response against AML-specific intracellular targets. (5) T cells expressing T-cell receptors (TCRs) engineered to recognize AML-specific intracellular antigens displayed by a Major Histocompatibility Complex (MHC) molecule.

**Figure 2 jcm-09-00554-f002:**
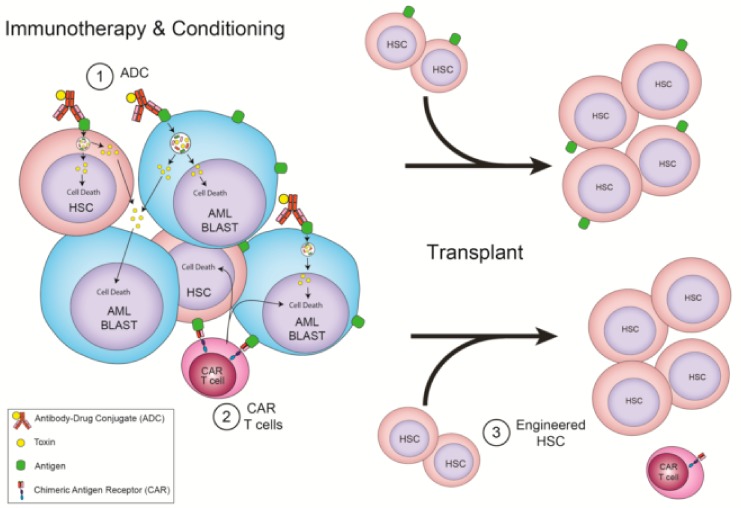
Immunotherapies that require allo-HSCT. (1) ADC that recognize HSC and AML shared antigens as conditioning for a transplant. (2) CAR T cells that recognize HSC and AML shared antigens and are depleted with conditioning for a transplant. (3) CAR T cells that recognize HSC and AML shared antigens as conditioning for a transplant, but whose donor cells have the target removed, allowing for CAR T cell persistence.

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
