# Peer review of "Is Hematopoietic Stem Cell Transplantation Required to Unleash the Full Potential of Immunotherapy in Acute Myeloid Leukemia?"

_jcm, 2020, doi:10.3390/jcm9020554_

Round 1

Reviewer 1 Report

This review is overally well organized and comprehensively introduces the current situation of AML immunotherapy and the dilemma of HSCT application.

Comments:

As two arms of the review: AML immunotherapy and HSCT, the former one is comprehensively introduced, while the latter one lacks more flavor. What are the current criteria of applying HSCT to treat AML patients? What is the standard procedure of transplanting HSCT? What are the pros and cons of HSCT for AML patients? What is the conditioning and how does it operate? What are the efficacies and differences between allogeneic and autologous HSCT on AML patients?

AML immunotherapy is still at its infancy, while CD19-directed CAR-T therapy is better documented in bridging it to allo-HSCT as a treatment. Better to introduce more on the current application and dispute of CD19 CAR-T with allo-HSCT as a background before digging into the AML immunotherapy with allo-HSCT.

Author Response

Reviewer 1

As two arms of the review: AML immunotherapy and HSCT, the former one is comprehensively introduced, while the latter one lacks more flavor. What are the current criteria of applying HSCT to treat AML patients? What is the standard procedure of transplanting HSCT? What are the pros and cons of HSCT for AML patients? What is the conditioning and how does it operate? What are the efficacies and differences between allogeneic and autologous HSCT on AML patients.

We appreciate that a more detailed introduction of allo-HSCT in AML would enhance the review and adding the following paragraph to provide context for the article.

“In this procedure, patients undergo conditioning chemotherapy with or without radiotherapy followed by transfusion of donor hematopoietic cells. Immunosuppression is required after the transplant to reduce the chance of Graft Versus Host Disease (GVHD). Allo-HSCT has been shown to reduce the relapse rate of AML and is the only potentially curative therapy in those with refractory disease [15,16]. The decision on whether to offer an allo-HSCT is complex and considers patient fitness, risk of AML relapse and availability of donors. Generally, if a suitable well-matched donor is available allo-HSCT is recommended for those with intermediate or adverse risk disease [15]. The major limitation is that most patients are not fit for the procedure due to their age at diagnosis and other co-morbidities. Autologous HSCT is an alternative to allo-HSCT that is associated with reduced toxicity, but is only effective in patients without high risk disease and is not widely utilized in all jurisdictions [17].”

What is the conditioning and how does it operate?

We have further described and introduced the role of conditioning with the following.
“Allo-HSCT requires chemotherapeutic agents as conditioning to suppress the hosts immune system while removing residual AML and HSPC. These factors are critical to allow for the safe engraftment of donor cells.”

AML immunotherapy is still at its infancy, while CD19-directed CAR-T therapy is better documented in bridging it to allo-HSCT as a treatment. Better to introduce more on the current
application and dispute of CD19 CAR-T with allo-HSCT as a background before digging into the AML immunotherapy with allo-HSCT.

We understand the reviewer’s point of view and have added the following section.

“The emergence of immunotherapies altering the treatment landscape in regard to allo-HSCT can be seen in the field of B-ALL. Anti-CD19 CAR T cells are capable inducing sustained Complete Remission (CR) thus potentially limiting the role of allo-HSCT in children and young adults with relapsed ALL [23]. Despite the emergence of CAR T cells in B-ALL it is unclear if the need for allo-HSCT will be reduced over time or if these therapies will play complementary roles. Given that relapse occurs post anti-CD19 CAR T the role of consolidation with an allo-HSCT is not currently established. [24,25].”

Reviewer 2 Report

In this work, Abadir et al review the different approaches to overcome the hematologic toxicity encountered during immunotherapy of AML with particular focus on stem  cell  transplantation. The article is well written and easy to read. However there are some minor points that will improve the quality of the manuscript:

Figura 1 does not clarify the main differences between transplant and no-transplant setting. I would suggest to change the figure and to clearly state the meaning of each symbol in the figure legend.

An additional figure illustrating the different strategies to mitigate hematologic toxicity would improve the manuscript substantially.

Author Response

Reviewer 2

Figure 1 does not clarify the main differences between transplant and no-transplant setting. I would suggest to change the figure and to clearly state the meaning of each symbol in the figure legend.
An additional figure illustrating the different strategies to mitigate hematologic toxicity would improve the manuscript substantially.

- As the reviewer suggested, we have improved the figures by splitting Figure 1 into two. The revised Figure 1 outlines the non HSCT strategies in greater detail with an improved legend. The revised Figure 2 outlines the strategies with HSCT again with an accompanying legend.
